# Qualitative and Quantitative Investigation of Multiple Large Eddy Simulation Aspects for Pollutant Dispersion in Street Canyons Using OpenFOAM

**Arsenios E. Chatzimichailidis [1]**, **Christos D. Argyropoulos [2]**, **Marc J. Assael [1]** and **Konstantinos E. Kakosimos [3],***

[1] Department of Chemical Engineering, Aristotle University of Thessaloniki, GR54636 Thessaloniki, Greece; archatzi@auth.gr (A.E.C.); assael@auth.gr (M.J.A.)
[2] System Reliability & Industrial Safety Lab, Institute of Nuclear & Radiological Sciences & Technology, Energy and Safety, National Centre for Scientific Research "Demokritos", GR15310 Athens, Greece; chris.argyropoulos@ipta.demokritos.gr
[3] Department of Chemical Engineering and Mary Kay O'Connor Process Safety Center, Texas A&M University at Qatar, Education City, PO BOX 23874 Doha, Qatar
* Correspondence: k.kakosimos@qatar.tamu.edu; Tel.: +974-442-30678

**Abstract:** Air pollution is probably the single largest environment risk to health and urban streets are the localized, relevant hotspots. Numerous studies reviewed the state-of-the-art models, proposed best-practice guidelines and explored, using various software, how different approaches (e.g., Reynolds-averaged Navier–Stokes (RANS), large eddy simulations (LES)) inter-compare. Open source tools are continuously attracting interest but lack of similar, extensive and comprehensive investigations. At the same time, their configuration varies significantly among the related studies leading to non-reproducible results. Therefore, the typical quasi-2D street canyon geometry was selected to employ the well-known open-source software OpenFOAM and to investigate and validate the main parameters affecting LES transient simulation of a pollutant dispersion. In brief, domain height slightly affected street level concentration but source height had a major impact. All sub-grid scale models predicted the velocity profiles adequately, but the k-equation SGS model best-resolved pollutant dispersion. Finally, an easily reproducible LES configuration is proposed that provided a satisfactory compromise between computational demands and accuracy.

**Keywords:** computational fluid dynamics; street canyon; atmospheric dispersion; large eddy simulation; turbulence modelling: subgrid-scale

## 1. Introduction

Air pollution is a well-known problem of modern cities and despite the great on pollutant control technology, it is still very difficult to ensure the elimination of the anticipated consequences in environment and human population. Therefore, the need for studying air pollution becomes apparent, since it is recognized as one of the main reasons of a large number of deaths [1]. Air pollution can be studied systematically by adopting various experimental and numerical methods, such as small air quality models (AQMs) [2], wind tunnel experiments [3], field measurements [4], computational fluid dynamics (CFD) models [5–7], and semi-empirical models [8,9].

Turbulence modelling is a core issue to address in CFD and it poses unique challenges for flows in the urban infrastructure [10]. The large size of buildings and high velocities of the atmospheric boundary layer result in highly turbulent flows [11], which are computationally expensive to

simulate [12]. Depending on the selected CFD technique, the whole spectrum of turbulent motions can either be resolved, modeled, or a combination of both.

Solutions of the Navier–Stokes equations for turbulent flows can be obtained using various numerical and analytical approaches, providing with different level of accuracy in each case. The most 'accurate' approach is the direct numerical simulation (DNS), which involves the numerical solution of the full spectrum of turbulence without the need of any modeling [13]. Even though DNS provides a high-fidelity representation of the physical phenomena, it is unfeasible for urban-scale flows and it is usually executed for flows with low Reynolds numbers owe to its high-computational cost [14]. On the other end, the Reynolds-averaged Navier–Stokes (RANS) simulation is a more feasible method that provides a time-averaged representation (or steady-state solution) of the studied phenomena with less demands in terms of computer speed and memory [15]. RANS (and unsteady-RANS) approach employs the Reynolds decomposition, a mathematical technique that separates the flow properties into the average and fluctuating components. This decomposition introduces the Reynolds-stress tensor, creating the well-known 'closure problem', which is solved by the addition of a turbulence model.

Another modelling approach, which demands less computational resources than DNS and resolves turbulence with more accuracy than unsteady-RANS, is the large eddy simulation (LES) method. LES begins with the filtering of Navier–Stokes equations, which divides turbulence into the large-scale resolved part and the sub-grid scale (SGS) modelled part [13]. The LES approach provides a time-dependent solution of the problem, with more details than RANS, as LES resolves a larger percentage of turbulence spectrum [16]. RANS and LES are the most common techniques to simulate atmospheric flows between buildings [17,18]. The general notion is that LES can capture the simulated phenomena with greater details, but it requires increasing computational power and time [19]. Due to their transient solution, LES can provide better prediction for the maximum values of concentrations, which is important for simulations involving toxic materials [20]. Moreover, LES is proven to provide results that are validated with experimental data from water channels [21,22], as well as wind tunnels [17,23–25]. Regarding the quasi-2D LES setup, there are many applied LES configurations. For example, Chung and Liu [26] employed the classic Smagorinsky SGS model [27], with the selected value of the Smagorinsky constant ($C_s$) being taken equal to 0.135. Moreover, Li studied the effect of the aspect ratio (*AR*) and heated walls to the flow and dispersion, for *AR* from 2 to 10 [21], using the one-equation SGS model by Moeng and Sullivan [28], which solves an additional transport equation for the SGS turbulence kinetic energy. At the solid boundaries, a 1/7 power-law wall model [29] was applied. The flow field was validated against water channel experimental data [30], while the concentration field was compared with wind tunnel measurements [31–33], presenting good agreement in all cases. Furthermore, Cai performed LES with the RAMS code [34] to investigate street canyon flow for quasi-2D [35] and 3D geometries [36]. The standard Smagorinsky model was adopted for the SGS modelling, with $C_s$ equal to 0.1. The value of the constant was further tested by Cui et al. [37] to show that larger $C_s$ values can cause excessive dissipation, while smaller ones create numerical errors. The obtained numerical results were compared with available wind tunnel experimental data [3,38,39] for the quasi-2D geometry [40] and water channel experimental data [41,42] for the 3D geometry.

Additionally, LES has been coupled with simplified and computationally affordable custom models of photochemical reactions under neutral atmospheric conditions in order to validate simple chemical AQMs [43]. For a deep street canyon (*AR* = 2), Zhong et al. [22] implemented the typical boundary conditions and compared the results with water channel data by Li et al. [30]. The k-equation eddy-viscosity SGS model [44] along with the logarithmic law of the rough wall [45] were used for the LES method. The obtained datasets of the spatial and temporal distributions of the reacting pollutants were used for validating a simple chemical box model [46].

Recently, Kikumoto and Ooka [47] performed a high resolution LES study for the investigation of turbulent dispersion in a unity street canyon, with the aim to reproduce a physical experiment of passive tracer dispersion. They predicted the mean and root-mean-square (RMS) velocity and

concentration levels, exhibiting fair agreement with the experimental data, except from slight underestimation for the concentration fluctuations near the source. The same work suggests that the LES method could eventually replace wind tunnel experiments, although with the burden of extremely high grid resolution.

In all these studies, the main objective was to simulate a street canyon flow using LES, while only in very few of them was there an attempt to describe and fully justify the appropriate LES configuration (e.g., SGS models, grid resolution, time-step and boundary conditions, among others) for the considered problem. To spend more time studying the employed phenomena, available CFD commercial codes are treated as black boxes and past works or guidelines are followed without adequate explanation or justification.

Therefore, in this study we selected a simple, yet important, street canyon geometry (described in Section 2) to conduct a comprehensive investigation, with verification and validation, of various parameters affecting LES. The open-source code OpenFOAM v5.0 [48] has been utilized for this investigation (details discussed in Section 3), for a typical quasi-2D geometry in 'real' scale, i.e., the height of buildings is set to 10 m. The investigated parameters are the domain height, the grid resolution, time step, Reynolds number, source height, four of the available SGS models, and three different wall functions. The obtained results for each parameter are qualitatively and quantitatively analyzed to understand its impact and identify the most suitable modelling configuration (presented in Section 4). Results are also compared extensively with available experimental and numerical data from the literature. Finally, the most important findings are summarized along with potential future studies (listed in Section 5).

## 2. Physical Problem

Many cities have complex layouts, created by a diverse variety of structures, ranging from the usual cubical buildings to skyscrapers and stadiums. Depending on the computational power and the scope of the work, the computational grid may contain whole cities, building blocks or street canyon(s), which is the basic structural unit of the city. Figure 1a presents the special (ideal case) of an 'infinite length' ($L$) street canyon, formed by two (rows of) rectangular parallelepiped buildings with height ($H$) and width ($B$), and a perpendicular wind direction (left to right). Because of the infinite length and the resulted symmetry, this is considered as a quasi-2D geometry.

Both the height and width of the buildings are set to 10 m, while the street width ($W$) can vary to depending on the studied $AR$ (here in $AR$ = 1). In the streamwise direction, the computational domain begins from the second half of the upwind and ends at the first half of the downwind building. In the spanwise direction, half ($H/2$) of the buildings' width is contained, effectively simulating a 'slice' of the street canyon. The upper boundary extends at five building heights ($5H$); more details in a following section.

Since the studied area is the smallest possible and the geometry is simple and ideal, the creation of a high resolution, fully structured, hexahedral mesh is easy and applicable. Structured hexahedral meshes discretize the computational domain more efficiently [49] and can have cells aligned with the flow direction, thus achieving smaller numerical errors [50] and better spatial convergence [51]. The height and width of the buildings are discretized with 120 and 60 cells, respectively, and the depth of the street is set to 60 cells, satisfying the demand for 3D LES. The cells are smaller near the solid boundaries, to resolve the flow characteristics of the wall boundary layer, reaching a 0.0038 m width. The Reynolds number ($Re = U_{ref} \cdot H/v$) is around 3,200,000, calculated using the reference velocity $U_{ref}$ at the top (5 m·s$^{-1}$), the kinematic viscosity of air, $v$, and the building height $H$.

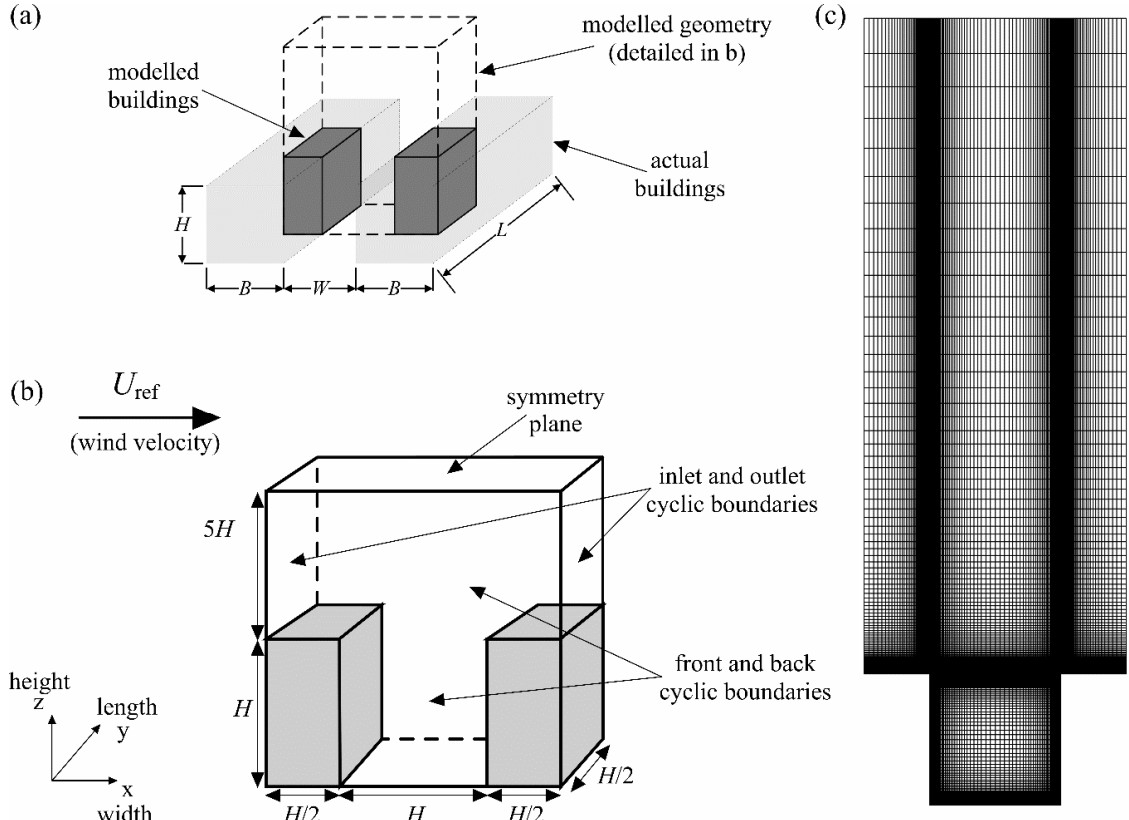

**Figure 1.** The quasi-2d street canyon geometry: (**a**) overview of the physical geometry and actual buildings (light shaded block), (**b**) detail of the modelled geometry and boundary conditions (discussed later); and (**c**) xz cross-section of a computational grid example.

## 3. Model Description

### 3.1. Governing Equations

The current study is carried out by using the LES methodology [52]. The unsteady, three-dimensional incompressible filtered Navier–Stokes equations [16] can be written in Cartesian tensor notation as follows.

Filtered continuity equation

$$\frac{\partial \overline{u}_i}{\partial x_i} = 0 \tag{1}$$

Filtered momentum equations

$$\frac{\partial \overline{u}_i}{\partial t} + \frac{\partial \overline{u}_i \overline{u}_j}{\partial x_j} = -\frac{1}{\rho}\frac{\partial \overline{p}}{\partial x_i} - \frac{\partial \tau_{ij}}{\partial x_j} + v\frac{\partial^2 \overline{u}_i}{\partial x_j \partial x_j} \tag{2}$$

where $\overline{u}_i$ is the filtered mean velocity component in the *i*-direction, $\overline{u}_j$ is the filtered mean velocity component in the *j*-direction, $x_{i,j}$ is the distance in the *i*- and *j*- directions, $\overline{p}$ is the filtered pressure, $v$ is the kinematic molecular viscosity, $\rho$ is the density, $t$ is the time and the subscript indices *i, j* (= 1, 2, 3) denote the three space coordinates. $\tau_{ij}$ is the SGS stress tensor and is analysed by the Leonard decomposition [53] as

$$\tau_{ij} = \overline{u_i u_j} - \overline{u}_i \overline{u}_j = L_{ij} + C_{ij} + R_{ij} = \left( \overline{\overline{u}_i \overline{u}_j} - \overline{u}_i \overline{u}_j \right) + \left( \overline{\overline{u}_i u'_j} + \overline{u'_i \overline{u}_j} \right) + \overline{u'_i u'_j} \tag{3}$$

where $L_{ij}$ is the Leonard term, $C_{ij}$ is the Cross term and $R_{ij}$ is the Reynolds term. Each term represents physical interactions between the resolved and modelled scales of the flow that arise from the LES approach [13].

The coupled pressure and velocity are solved with the PISO (Pressure-Implicit with Splitting of Operators) algorithm by Issa [54]. The pressure equation is treated implicitly by the *PCG* solver using the *DIC* preconditioner, while the momentum equation is resolved by a simple solver, namely *smoothsolver* in OpenFOAM, along with the *GaussSeidel* smoother. The terms of the equations are discretized with second-order numerical approximations, according to the standard practice [55]. The time first derivative is discretized with the backward difference scheme (*backward* in OpenFOAM), while the UMIST scheme [56] is employed for the convective terms. The gradient and the Laplacian terms are discretized using the central difference schemes, *Gauss linear* and *Gauss linear corrected* [48], respectively.

The sub-grid part of turbulence is represented by the SGS stress tensor ($\tau_{ij}$) and can be modelled with various methods. A common approach is the Boussinesq approximation, which assumes that the deviatoric part of the SGS tensor is proportional to the resolved strain rate tensor $\overline{S}_{ij}$, through the eddy viscosity, $v_{\text{sgs}}$,

$$\tau_{ij}^d = \tau_{ij} - \frac{1}{3}\tau_{kk}\delta_{ij} = -2v_{\text{sgs}}\overline{S}_{ij} \tag{4}$$

where $\tau_{ij}^d$ is the deviatoric part, $\tau_{ij}$ is the normal component and $\tau_{kk}$ is the shear component of the SGS stress tensor. $\delta_{ij}$ is the Kronecker delta and $\frac{1}{3}\tau_{kk}\delta_{ij}$ is the hydrostatic tensor, which is embedded in the modified filtered pressure term. The resolved strain rate tensor $\overline{S}_{ij}$ is written as

$$\overline{S}_{ij} = \frac{1}{2}\left(\frac{\partial \overline{u}_i}{\partial x_j} + \frac{\partial \overline{u}_j}{\partial x_i}\right) \tag{5}$$

Doing so, the problem of unresolved modelled turbulence has been transformed to the calculation of eddy viscosity $v_{\text{sgs}}$. In the present study, the considered SGS models are presented in Section 3.2.

Finally, the selected incompressible solver *pisoFoam* [48] was modified to solve the convection–diffusion equation for a passive scalar, at every time-step.

Convection—diffusion equation

$$\frac{\partial C}{\partial t} = \nabla^2(D_{\text{total}}C) - \nabla \cdot (\overline{u}_i C) + P \tag{6}$$

where $C$ is the concentration, $P$ are the characteristics of the pollutant source (i.e., geometry and emission rate) and $D_{\text{total}}$ is the total diffusion coefficient

$$D_{\text{total}} = D + \frac{v_{\text{sgs}}}{Sc_{\text{t}}} \tag{7}$$

where $D$ is the molecular diffusion coefficient, $v_{\text{sgs}}$ is the eddy viscosity, and $Sc_{\text{t}}$ is the turbulent Schmidt number. The value of $Sc_{\text{t}}$ is taken equal to 0.7, in accordance with previous atmospheric dispersion studies [22,57], while $D$ was equal to $1.64 \cdot 10^{-5}$ m$^2 \cdot$s$^{-1}$, which is the molecular diffusion coefficient for, a typical pollutant, $CO_2$ at 25 °C [58].

*3.2. SGS Modeling*

3.2.1. Standard Smagorinsky Model

The standard Smagorinsky model [27] is the most well-known SGS model and constitutes the base for the development of many other SGS models. The main equation of the model based on the eddy-viscosity hypothesis can be written as

$$\nu_{\text{sgs}} = (C_{\text{s}}\Delta)^2 |\overline{S}| \tag{8}$$

where $C_{\text{s}}$ is the Smagorinsky coefficient, $|S| = \sqrt{2\overline{S}_{ij}\overline{S}_{ij}}$ is the magnitude of the resolved strain rate tensor and $\Delta$ is the filter width, usually associated with the cell dimensions $\Delta_i$, $\Delta = (\Delta_x\Delta_y\Delta_z)^{1/3}$.

The behavior of the standard Smagorinsky depends heavily on its constant, $C_{\text{s}}$, and suffers from many weaknesses (e.g., backscatter phenomenon, tuning of $C_{\text{s}}$ and requirement of a damping function). The model also fails to eliminate the eddy viscosity near to the wall boundaries and the requirement of the van Driest damping function [59] is necessary to reduce the $C_{\text{s}}$ value to zero and ensure the no-slip boundary condition.

3.2.2. WALE Model

The variations of the Smagorinsky model calculate the SGS turbulence, based on its connection with the symmetric part of the velocity gradient tensor, i.e., the resolved strain rate tensor, effectively capturing the vortex deformation. The local strain and rotation rates of the smallest resolved turbulence scales are taking into account in the WALE model by Ducros et al. [60]. It is reported that WALE reproduces the transition from laminar to turbulent flow and the correct scaling of turbulence close to the walls without the requirement of a damping function [61]. According to the WALE model the eddy viscosity is given by

$$\nu_{\text{sgs}} = (C_{\text{w}}\Delta)^2 \frac{\left(S_{ij}^d S_{ij}^d\right)^{3/2}}{\left(\overline{S}_{ij}\overline{S}_{ij}\right)^{5/2} + \left(S_{ij}^d S_{ij}^d\right)^{5/4}} \tag{9}$$

where $\Delta$ is the usual grid length and $C_{\text{w}} = 0.325$ is the WALE constant. $S_{ij}^d$ is the deviatoric symmetric part of the square for the resolved velocity gradient tensor and is defined as

$$S_{ij}^d = \frac{1}{2}\left(\overline{g}_{ij}^2 + \overline{g}_{ji}^2\right) - \frac{1}{3}\delta_{ij}\overline{g}_{kk}^2 \tag{10}$$

where $\overline{g}_{ij} = \partial\overline{u}_i/\partial x_j$ are the velocity gradients.

3.2.3. k-Equation Model

The above-mentioned algebraic models connect $\nu_{\text{sgs}}$ to the local instantaneous resolved velocity gradients. On the other hand, there are regions with low resolved gradients, but high SGS kinetic energy because of convection from regions with high sub-grid scale turbulence [62]. A common approach to capture the historic effects of production, dissipation, and diffusion of the SGS kinetic energy ($k_{\text{sgs}}$) is the resolving of its transport equation. The k-equation model by Yoshizawa and Horiuti [63] resolves the transport equation of $k_{\text{sgs}} = \frac{1}{2}\left(\overline{u_k u_k} - \overline{u}_k\overline{u}_k\right)$ leads to

$$\frac{\partial k_{\text{sgs}}}{\partial t} + \frac{\partial k_{\text{sgs}}\overline{u}_j}{\partial x_j} = \frac{\partial}{\partial x_j}\left[(\nu + \nu_{\text{sgs}})\frac{\partial k_{\text{sgs}}}{\partial x_j}\right] - \tau_{ij}\overline{S}_{ij} - C_{\text{e}}\frac{k_{\text{sgs}}^{3/2}}{\Delta} \tag{11}$$

and the eddy viscosity is given by

$$\nu_{\text{sgs}} = C_{\text{k}}k_{\text{sgs}}^{1/2}\Delta \tag{12}$$

where $v$ is the laminar viscosity, $C_k$ (=0.094) and $C_e$ (=1.048) are the model constants. The terms in the right hand of Equation (11) represent the diffusion, production and dissipation of $k_{sgs}$, respectively. The constants $C_k$ and $C_e$ are calculated with the assumption of balance between SGS energy production and dissipation rates [64], similar to the standard Smagorinsky model, presenting the same limitations. Consequently, improved versions of the k-equation model have been proposed to calculate the constants dynamically, depending on the local turbulence.

### 3.2.4. Dynamic k-Equation Model

The localised dynamic k-equation model by Kim and Menon [65] employs a similar operation to that introduced by Lilly [66] for the dynamic Smagorinsky model. A second test filter $\hat{\Delta} = 2\overline{\Delta}$ and the Germano identity are used to derive equations that relate the values of $C_k$ and $C_e$ to quantities resolved at the test filter scale [67]

$$C_{k,\,\text{dynamic}} = \frac{1}{2}\frac{L_{ij}\sigma_{ij}}{\sigma_{ij}\sigma_{ij}} \tag{13}$$

where

$$\sigma_{ij} = -k_{\text{test}}^{1/2}\overline{S}_{ij}\hat{\Delta} \tag{14}$$

$$k_{\text{test}} = \frac{1}{2}\left(\widehat{\overline{u_k u_k}} - \hat{\overline{u}}_k\hat{\overline{u}}_k\right) \tag{15}$$

and

$$C_{e,\,\text{dynamic}} = \frac{(v + v_{\text{sgs}})\left(\frac{\partial\overline{u}_i}{\partial x_j}\widehat{\frac{\partial\overline{u}_i}{\partial x_j}} - \frac{\partial\hat{\overline{u}}_i}{\partial x_j}\frac{\partial\hat{\overline{u}}_i}{\partial x_j}\right)\hat{\Delta}}{k_{\text{test}}^{3/2}} \tag{16}$$

### 3.3. Boundary Conditions

In the streamwise direction, periodic boundary conditions were imposed (*cyclic* in OpenFOAM) for all the flow variables, i.e., $U$, $p$, $v_{\text{sgs}}$, and $C$. The main flow was recycled from the outlet boundary patch to the inlet, ensuring that the boundary conditions are time-dependent, a requisite for LES [68]. To initialize the turbulence, a single velocity vector with the value of the reference velocity ($5\ \text{m}\cdot\text{s}^{-1}$) was imposed to the whole computational grid. The simulation of the wind flow begun and gradually the turbulence inside the street canyon dissipated forming the expected recirculating vortex, while the turbulence in the upper atmosphere created a realistic wind profile. The simulation ran for a sufficient time span, in which the obtained average velocity was statistically correct, when compared with experimental data and the flow was judged as realistic (i.e., the instantaneous snapshots of velocity showed no indications of laminar flow). Only after that, the emission of the pollutants from the source begun. This method is well-established in similar works, e.g., [21,22,37]. The recycling of the concentration was countered by overwriting $C$ at the inlet patch, at every time-step with the utility *scalarFixedValueConstraint*, available after OpenFOAM 3.0.

Another option for the streamwise boundaries could be the *mapped* boundary conditions, in which the user can choose which parameters will be recycled. These conditions were used extensively by the authors in OpenFOAM v2.3.1, with the results of flow being statistically valid for $AR$ = 1 and 0.5. The method was also used successfully for $AR$ = 0.33, but for $AR$ = 0.25 a recirculating flow was found to develop just after the inlet patch above the upwind building. As solution progressed, this flow was amplified, eventually causing the simulation to diverge. The reason for this was not investigated further neither this method was tested in newer versions. Therefore, cyclic conditions have been employed which proved more stable in either versions of OpenFOAM, v2.3.1 and v5.0. At the top of the domain, the symmetry boundary condition (*symmetryPlane* in OpenFOAM) was set, following the common practice for the quasi-2D setup, e.g., [22,40,69]. This means that the normal velocity on this patch is assigned to zero and for the scalars there is a zero-normal gradient.

Periodic conditions are used also for the spanwise direction, again as the common practice for the quasi-2D configuration. The recycling of the random small velocity vectors from patch to patch is not

affecting the main flow. Furthermore, tests with symmetry for these patches resulted in unnatural flows. The wall boundaries were set to no-slip condition for the velocity field (*fixedValue* in OpenFOAM)

$$\overline{U}_i = 0;\ \overline{U}_j = 0;\ \overline{U}_k = 0 \tag{17}$$

where $\overline{U}$ is the filtered velocity component and $i$, $j$ and $k$ are the three x, y and z-coordinates, respectively. The fixed gradient (*zeroGradient* in OpenFOAM) was used for the scalar parameters.

$$\frac{\partial \varphi}{\partial x_i} = 0;\ \frac{\partial \varphi}{\partial x_j} = 0;\ \frac{\partial \varphi}{\partial x_k} = 0 \tag{18}$$

where $\varphi$ is the scalar quantity, i.e., $p$, $v_{\text{sgs}}$, and $C$.

For high *Re* flows or relatively coarse grids, depending on the non-dimensional distance from the wall to the first grid point ($y^+$), special treatment should be employed for the flow characteristics on the wall boundaries [13]. OpenFOAM provides a variety of wall functions to handle flow characteristics on walls [70]. Herein, two functions studied combined with the standard Smagorinsky model to calculate the value of turbulent viscosity $v_{\text{sgs}}$ on the walls. The equation that relates $v_{\text{sgs}}$ to $y^+$ is

$$v_{\text{sgs}} = v \left( \frac{\kappa y^+}{\ln(E y^+)} - 1 \right) \tag{19}$$

where $\kappa = 0.41$ is the von Karman's constant and $E = 9.8$ is a constant that describes the velocity profile near the wall.

The well-known wall function by Launder and Spalding [71] is implemented in OpenFOAM v5.0 as *nutUSpaldingWallFunction*, for which $y^+$ is calculated as a function of the dimensionless velocity $u^+$

$$y^+ = u^+ + \frac{1}{E} \left[ e^{\kappa u^+} - 1 - \kappa u^+ - \frac{1}{2} \left( \kappa u^+ \right)^2 - \frac{1}{6} \left( \kappa u^+ \right)^3 \right] \tag{20}$$

A simple logarithmic wall function (*nutkWallFunction*) was also tested, that estimates $v_{\text{sgs}}$ using Equation (20) for values of $y^+ > 11$, or else assumes $v_{\text{sgs}} = 0$.

### 3.4. Computational Details

The solution of LES provides dynamic time-series, which are usually averaged over a feasible and representative time-span. Previous works have specified this time-span, as a function of $U_{\text{ref}}$ and a characteristic length. For example, Li et al. [69] used 400 $H/U_{\text{ref}}$ to reach a "pseudo-steady state" and another 300 $H/U_{\text{ref}}$ to collect the appropriate statistics. Employing a similar logic for turbulence generation, Boppana et al. [72] used the streamwise length of the domain $L_x$ to determine the respective time-spans as 120 and 200 $L_x/U_{\text{ref}}$. For this work, the fully developed flow field was obtained after 150 $H/U_{\text{ref}}$, while the statistics were collected over 650 $H/U_{\text{ref}}$ (or 325 $L_x/U_{\text{ref}}$, or 1300 s). The computational cost for these calculations requires the employment of high-performance computing (HPC). Indicatively, the fine grid required a computational time of three months using 240 CPU cores, which for the medium grid dropped to seventeen days, this time employing 192 CPU cores.

### 4. Results and Discussion

This section presents the detailed results of the investigation along with the qualitative analysis and quantitative comparison with literature data. The results relay to average and instantaneous values of the velocity and to averaged concentrations, for the wind flow and pollutant dispersion, respectively. At first, the investigation is based on a standard model configuration employing the Standard Smagorinsky model and the van Driest wall function, over a coarse resolution grid (80-cells across the street canyon; 768,000 in total) and a time-step of $1 \times 10^{-3}$ s. As the investigation progresses, the model configuration is updated.

Both experimental and numerical literature data are used to compare with own results. For the velocity components the experimental data by Li et al. [30] and the numerical data (also LES) by Li [73] are employed. For the pollutant concentrations the experimental data by Pavageau and Schatzmann [31], Meroney et al. [32], Pavageau [33], and Hoydysh and Dabberdt [74] are employed. The quantitative comparison follows well-known metrics (i.e., *FAC2*, *FB*, and *NMSE)* proposed by collaborative projects such as the COST Action 732 [75], according to the works of Hanna [76] and Chang and Hanna [77], and recently for the validation of LES, e.g., [25,78,79].

*4.1. Domain Height*

Best practice guidelines suggest that the external boundary patches should be far away from the investigated area, typically around five times the average building height [55]. In similar studies, this area has been taken equal to 1*H* [80], 2*H* [22], 3*H* [69], or 4*H* [81] building heights. This dimension controls the available space for the wind flow (velocity profile) to be developed. It is important to be large enough, so the profile can resemble the 'stratification' of the urban boundary layer. The first option for a larger upper atmosphere height is to increase the number of cells proportionally, which consequently increases the computational cost for the less studied part of the flow. The second option is to keep the same number of cells and adjust the cell-to-cell ratio, so that the cells at the roof level inside the street canyon and the upper atmosphere match the same size. Here, the second option was selected. Also, note that the inlet, outlet, front, and back patches did not follow this recommendation owe to the cyclic boundary conditions.

Therefore, a sensitivity analysis was conducted using 2*H*, 3*H*, 4*H*, and 5*H* to determine the appropriate height. The grid resolution was kept at 80 cells per building height to compromise between accuracy and computational cost for the 2*H*–5*H*, however, a medium (120 cells) and a fine (180 cells) grid were also examined for 5*H*. More details on the grid resolution are presented in the next section. Figure 2 presents the average concentrations (normalised by $C_{max}$) on the two walls of the unity street canyon compared with experimental data by Pavageau and Schatzmann [31], Meroney et al. [32], Pavageau [33], and Hoydysh and Dabberdt [74]. The LES results by Li [73] are also exhibited for comparison.

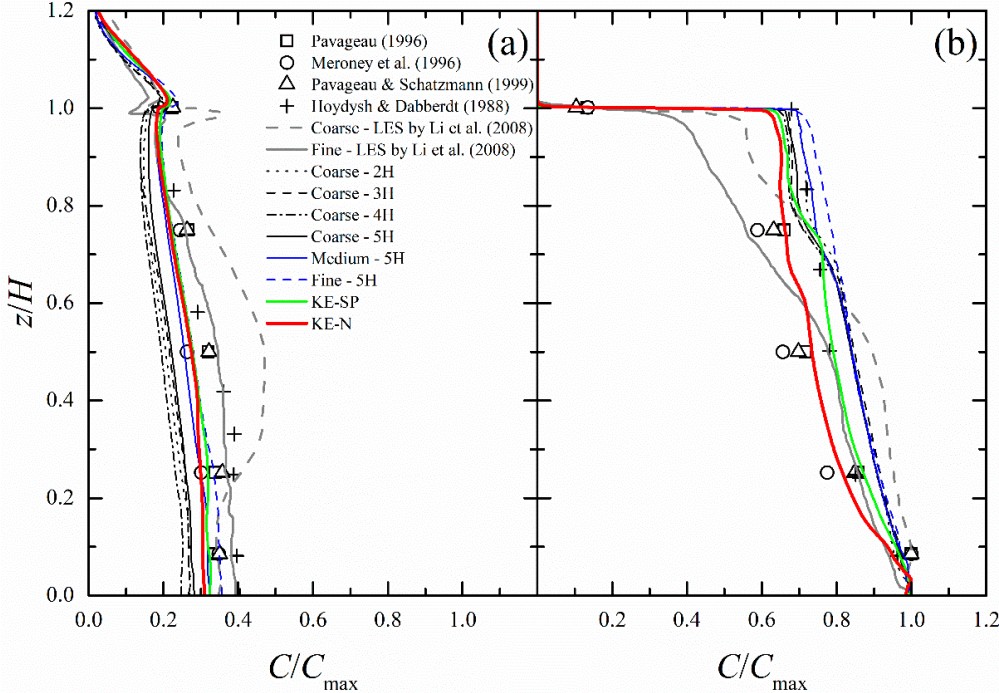

**Figure 2.** Vertical dimensionless concentration profiles for the upper atmosphere height and grid resolution: (**a**) the downwind and (**b**) the upwind wall for the unity street canyon.

The concentration results for the upwind wall are in fair agreement with the experimental data and the obtained values for the four upper atmosphere heights are nearly identical, while the largest variation is observed at the upper part of the canyon. For the downwind wall, the results for 2*H*, 3*H*, and 4*H* are different among them, while for 5*H* are marginally closer to the experimental data. The results for the downwind wall are affected more than those for the upwind side, as this side is characterized by the entrainment of clean air from the upper atmosphere, which is in turn controlled by the wind velocity profile. For example, the transition of the flow from the top of the domain to the shear layer for the 2*H* is more abrupt and unnatural than that of 5*H*. On the other hand, the upwind side is influenced by the phenomena developed inside the canyon, mainly the recirculating vortex that dominates the flow. It is concluded that the experimental data are replicated better in the 5*H* case due to the larger available area which allows for a less-constrained development of the wind flow.

### 4.2. Grid Resolution

A grid sensitivity analysis was conducted to quantify the error induced by discretization, utilizing the grid convergence index (*GCI*) [82] and the LES index of quality (or *LES_IQ*) [83].

The *GCI* estimates an error in a 'loose statistical sense' for measuring the effect of a change on the grid resolution, i.e., the refinement/transition from one grid to another and can be calculated as

$$GCI_{\text{fine}} = F_s \frac{|e_{21}|}{r_{21}^p - 1}; \; e_{21} = f_2 - f_1 \tag{21}$$

where $F_s$ is the safety factor, $r$ is the refinement ratio, and $p$ is the order accuracy of the used method, while $f_1$ and $f_2$ are the solutions from each used grid. Three grids with increasing density, i.e., a constant refinement ratio of 1.5 with $F_s = 1.25$, were tested following the common practice [12,20,84,85]: a coarse grid of 768,000 cells ($\Delta x = \Delta z = 0.125$ m, $\Delta y = 0.063$ m), a medium grid of 2,592,000 cells ($\Delta x = \Delta z = 0.083$ m, $\Delta y = 0.042$ m) and a fine grid of 8,748,000 cells ($\Delta x = \Delta z = 0.056$ m, $\Delta y = 0.028$ m). Note that the adopted numerical schemes for the discretization are second order accurate ($p = 2$).

The grid-induced error was inspected at three areas, the relatively low-resolution area in the middle of the canyon and the denser upwind and downwind areas, where the polluted air removal and clean air entrainment occur, respectively. Thus, the *GCI* was calculated in the middle vertical plane of the grid ($y/L = 0.5$) for three profiles across the canyon-width at $x/W = 0.25$, 0.50, and 0.75, respectively. The $GCI_{\text{fine}}$ values for the medium to fine grid refinement are presented in Figure 3.

The $GCI_{\text{fine}}$ values are proportionally larger for $U_z/U_{\text{ref}}$ than $U_x/U_{\text{ref}}$, although the absolute values are smaller for the first. The larger differences for $U_z/U_{\text{ref}}$ are observed in the middle height of the canyon, where the grid is relatively coarse, as shown in Figure 3d,f. On the other hand, for $U_x/U_{\text{ref}}$ the lower $GCI_{\text{fine}}$ values are calculated inside the canyon area, while abnormally high values are identified in the area above the buildings (Figure 3a–c), although the velocity at the top of the domain is virtually the same (~5.1 m/s) for all the grids. This could be linked with the observation that the velocity profile for the fine grid adapts more abruptly towards the street canyon.

The $GCI_{\text{medium}}$ (not presented here; for the coarse to medium grid refinement) was found to be in all cases larger than that of the fine grid. For example, the average $GCI_{\text{medium}}$ for the $U_z/U_{\text{ref}}$ inside the canyon was calculated as 1.8, 0.5, and 1.1% for the three vertical profiles, while the values for $GCI_{\text{fine}}$ were found to be 0.6, 0.3, and 1%, respectively. This trend is also valid for the $U_x/U_{\text{ref}}$, with the values for $GCI_{\text{medium}}$ were found to be equal to 1.1, 1.9, and 1.3% for the three vertical lines, while the values for $GCI_{\text{fine}}$ were recorded as 0.7, 1, and 0.7%, respectively. As expected, the transition from the medium to fine grid gives smaller differences than the transition from the coarse to the medium. Nevertheless, the $GCI_{\text{fine}}$ is less than 1%, implying that the medium grid is sufficient for the wind flow solution.

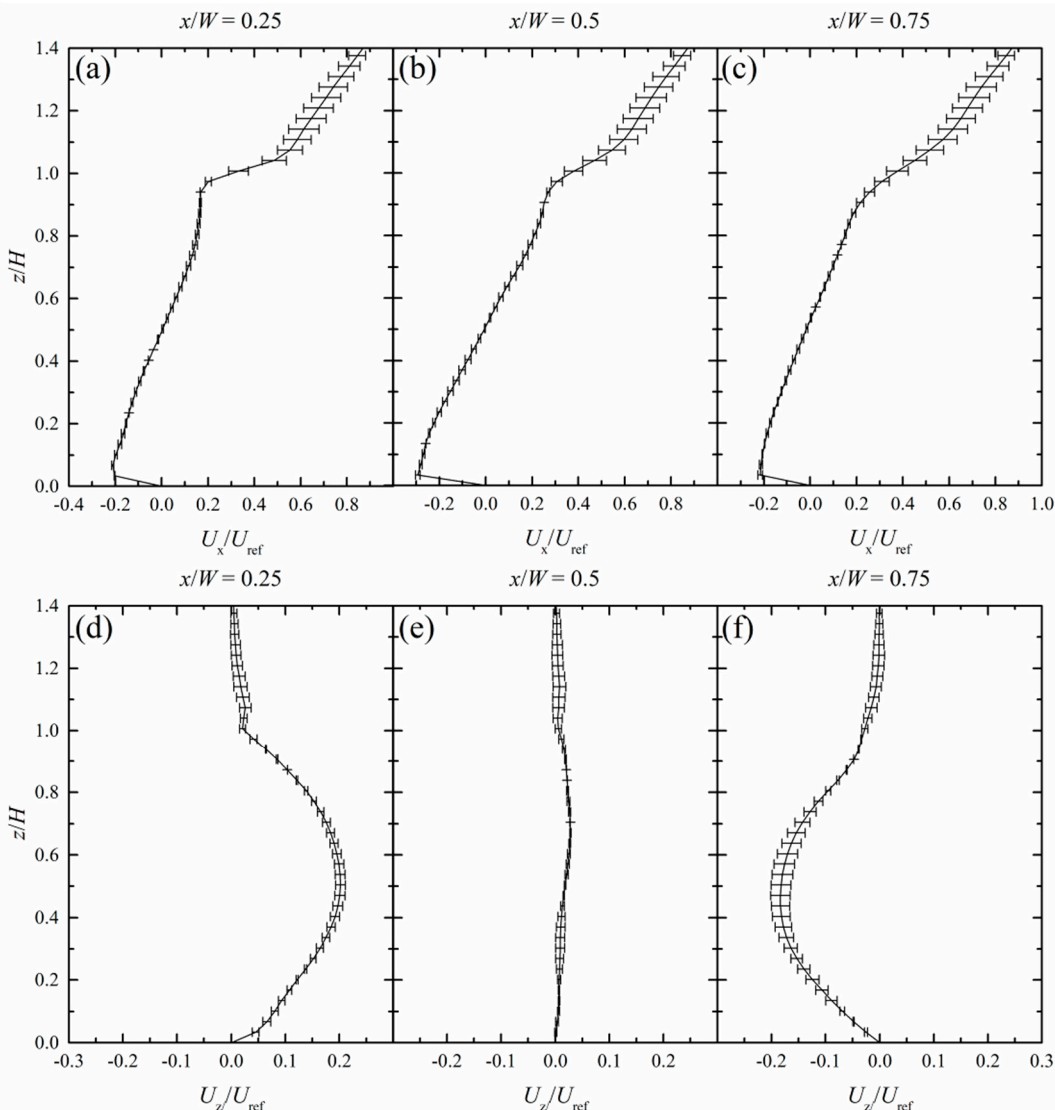

**Figure 3.** The calculated GCI$_{\text{fine}}$ for the medium to fine grid refinement with the corresponding dimensionless $U_{\text{x}}$ and $U_{\text{z}}$ at x/W = 0.25, x/W = 0.50, and x/W = 0.75.

The average concentration profiles of the numerical and experimental data for the two canyon walls are compared in Figure 2. As the grid resolution increases, the concentration of the downwind wall approaches the experimental and numerical values. The best results obtained from the fine grid, although the results from the medium grid are also in good agreement with the experiments. For the upwind wall and from the ground up to $z/H = 0.6$, the three solutions are identical and larger than the experimental results, while they differ for the distance until the roof height. For the part from $z/H = 0.7$ up to the roof, the results of the coarse grid are closer to the experimental, while the medium grid exhibits reasonable agreement. The results for GCI for $C/C_{\text{max}}$ are presented in Figure 4.

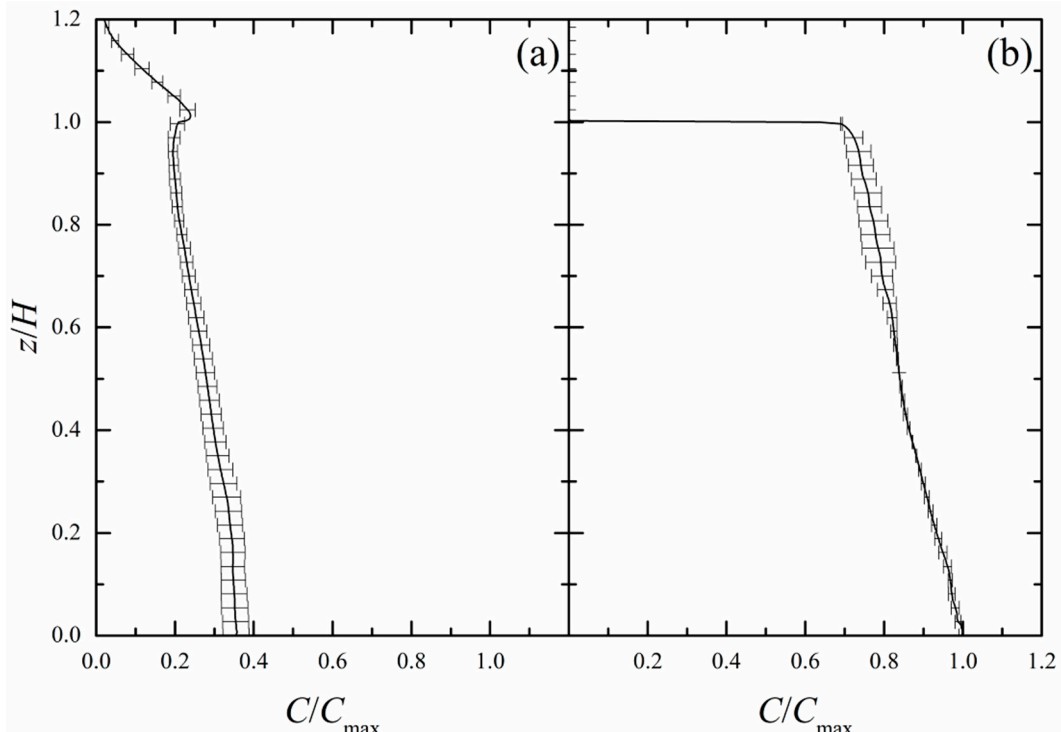

**Figure 4.** Calculated $GCI_{\text{fine}}$ for the medium to fine grid refinement and corresponding averaged and dimensionless concentration $C/C_{\text{max}}$ at (**a**) the downwind and (**b**) the upwind walls.

For $C/C_{\text{max}}$, the average $GCI_{\text{medium}}$ was calculated as 0.8% and 3.1% for the upwind and the downwind walls, respectively. The $GCI_{\text{fine}}$ was reduced even more for the downwind wall at 2.2%, but was increased for the upwind wall at 1.2%. Nevertheless, this discrepancy reveals the drawbacks of *GCI*, as the transition from medium to fine just moved the results closer to the experimental, as shown in Figure 2.

While, the use of *GCI* determined that the uncertainty induced by grid is low enough, other metrics are required to find out the percentage of the resolved turbulence. According to Pope [86], LES is properly employed, when 80% of the turbulence kinetic energy is resolved, a value that will be examined using the LES index of quality (or *LES_IQ*) by Celik et al. [83], with the equation involving the laminar $v$ and turbulent viscosity $v_{\text{sgs}}$

$$LES\_IQ_v = \frac{k_{\text{resolved}}}{k_{\text{total}}} = \frac{1}{1 + 0.05\left(\frac{v + v_{sgs}}{v}\right)^{0.53}} \tag{22}$$

In Figure 5, it is observed that a region of relatively lower $LES\_IQ_v$ exists in front of the downwind building for the selected three grids. This is the area where occurs the entrainment of clean air and is characterized by intensive turbulence. Therefore, very coarse grids may have difficulties to resolve the turbulence kinetic energy at the acceptable 80% for this area, which does not occur for the selected grids. Another area of intense turbulence is the shear layer between the roofs, where $LES\_IQ_v$ exceeds 85% for all resolutions, due to the dense grid. Outside the street canyon, $LES\_IQ_v$ has significantly lower values than inside the canyon, reaching 70% for the coarse grid in Figure 5a. Similarly, in the same area, the medium and the fine grid in Figure 5b,c present their lower values at 75% and 80%, respectively. On the other hand, this should not affect the quality when using the medium grid, because the main area of interest is inside street canyon.

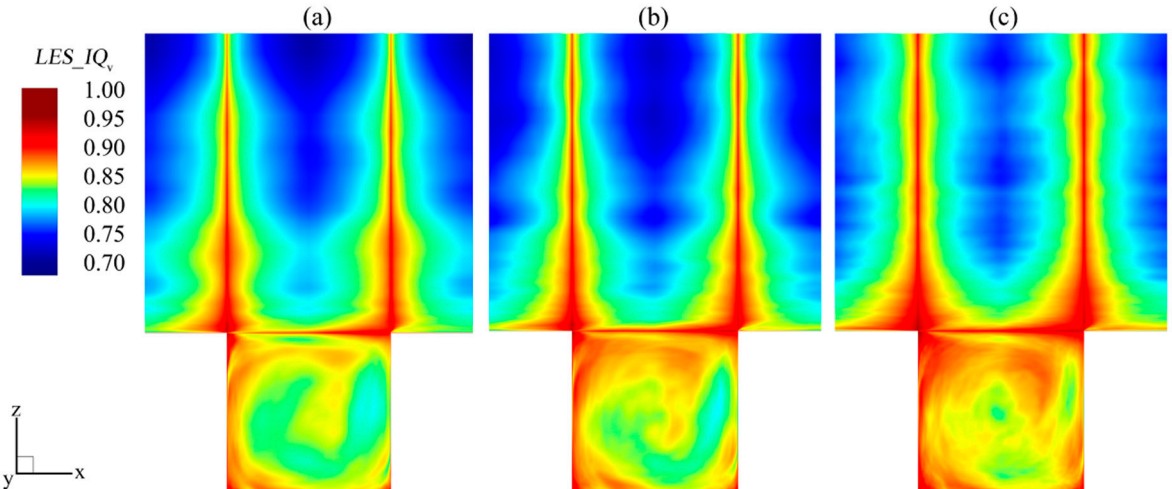

**Figure 5.** The *LES_IQ*$_\text{v}$ index for three selected grids: (**a**) coarse, (**b**) medium, and (**c**) fine.

For the coarse grid in Figure 5a, the *LES_IQ*$_\text{v}$ is higher than 85% inside the canyon, while the lower values around 80% are in the outside area. As for the medium grid in Figure 5b, the turbulence is resolved at more than 85% inside the street canyon, while the percentage is over 90% for the walls and the shear layer. The value of *LES_IQ*$_\text{v}$ is lower than the 80% (lower limit) at the outside area and approximately at one building height above the roofs. Again, this should be acceptable because turbulence is resolved at the areas that matter the most for this study. Finally, inside the street canyon the fine grid demonstrates values higher than 88%, while the values close to the wall reach to 95%, as shown in Figure 5c.

The use of *LES_IQ*$_\text{v}$ provides valuable insights about the resolving of turbulence in the relatively high resolutions, usually utilized for the quasi-2D setup. The resolution of the coarse grid proved not sufficient for an LES study at the *Re* of 3,200,000. The medium grid performed satisfactory for the street canyon area, while showed some deficiencies for the upper region. The fine grid provided excellent compliance with the 80% limit for both areas. Nevertheless, the medium grid is judged to be a fair compromise between the required computational time and level of resolving turbulence.

### 4.3. Time-Step

Along with the spatial discretization, the selection of the appropriate time-step ($\Delta t$) must also ensure that the evolution of the small vortex structures will be described [68]. Employing a structured grid generally leads to smaller time-step requirements. The small cells on the building walls combined with the hexahedral structured grid result in unnecessarily very small cells outside the street canyon, where velocity reaches its maximum. In turn, very small time-steps are required to maintain reasonably low Courant–Friedrichs–Lewy (*CFL*) number throughout the domain. A typical example of this situation can be found at Figure S1 of the supplementary material.

In this study, for the medium grid and $U_\text{ref}$ = 5 m·s$^{-1}$, three different $\Delta t$ values were examined, namely $6 \times 10^{-4}$ s, $8 \times 10^{-4}$ s, and $9 \times 10^{-4}$ s. Any value larger than $9 \times 10^{-4}$ s caused numerical instabilities, leading to diverging solutions.

Following the works of Kornhass et al. [87] and Ai and Mak [88], the obtained velocity time-series for the three time-steps were transformed to the frequency domain using the fast Fourier transformation (FFT) technique. Figure 6 presents the spectra of turbulence energy generated for the $U_\text{x}$ and $U_\text{z}$ velocity components, for a point in the upwind side with $x/W$ = 0.1 and $z/H$ = 0.9.

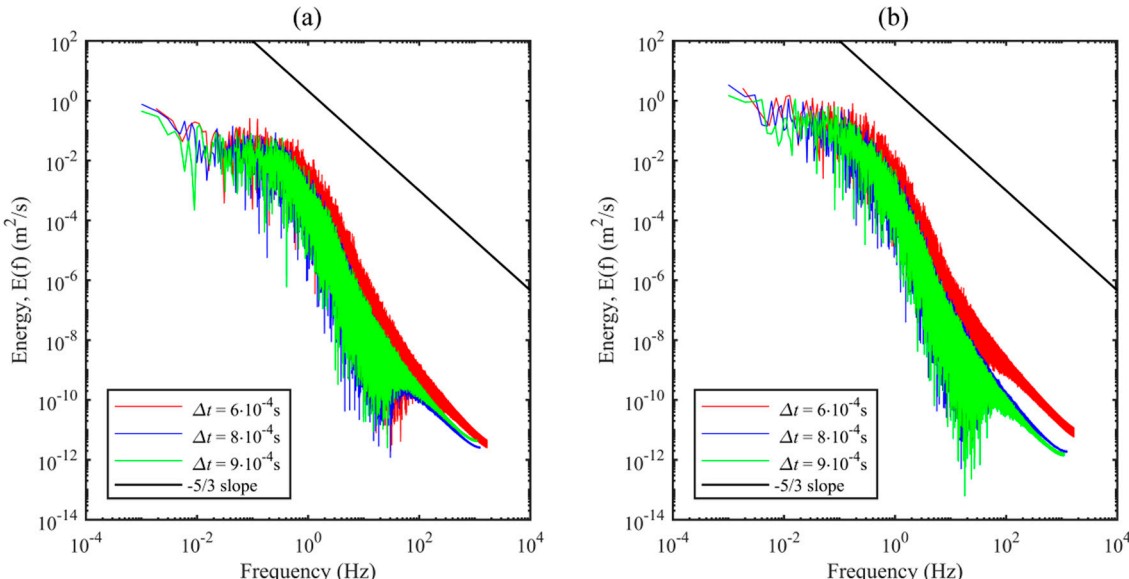

**Figure 6.** Comparison of the turbulence energy spectra in the frequency domain of velocity: (**a**) $U_x$ and (**b**) $U_z$ at $x/W = 0.1$ and $z/H = 0.9$ on the upwind side of the street canyon. The solid red time series represent the $\Delta t = 6 \times 10^{-4}$ s, blue the $\Delta t = 8 \times 10^{-4}$ s and green the $\Delta t = 9 \times 10^{-4}$ s.

For both velocity components in Figure 6a,b, the frequencies present only small differences in their minimum and maximum values for the three $\Delta t$ values. This is expected, as the values of the $\Delta t$ are close to each other. Six frequency scales from $10^{-3}$ to $10^3$ Hz are covered, reaching the significant high frequencies that represent the small-scale turbulence. Furthermore, the inertial sub-range of turbulence is covered between 1 and 10 Hz for both $U_x$ and $U_z$, while turbulence is shown to dissipate faster for larger frequencies.

In Figure 6a, the power amplitudes for $U_x$ exhibit small differences among the three selected $\Delta t$ values. The coverage of the high frequencies occurs slightly more for the smaller $\Delta t$ (2000 Hz), while the difference for the medium and large $\Delta t$ is very small (1600 Hz instead of 1400 Hz). The same maximum frequencies are observed for $U_z$ in Figure 6b. On the other hand, the calculated power amplitude is slightly larger for the smaller $\Delta t = 6 \times 10^{-4}$ s, while the result for the medium and large $\Delta t$ is the same.

For all cases of $\Delta t$ and velocity components, the turbulence spectra present a coverage of the inertial sub-range of turbulence, up to higher frequencies in the dissipation range. As a fair compromise between the coverage of the small-scale turbulence and computational cost, the medium $\Delta t = 6 \times 10^{-4}$ s is selected for the rest of the study.

*4.4. Reynolds Number*

The high values of wind velocity and building height result into high Reynolds numbers (*Re*) of atmospheric flows. Wind tunnel studies by Snyder [89] set the critical *Re* number at 11,000, based on the reference velocity of the free stream and building height. Later experiments either followed the aforementioned critical number [90], or directed to larger *Re* numbers, e.g., 19,000 [17], 37,000 [91], 44,000 [92], and 56,000 [3]. Recently Cui et al. [93] reviewed the latest *Re* values reported in the literature and available CFD simulations with regard to the critical *Re* number, proposing a critical value of 32,000. At the same time, following such critical values do not result to validation issues since most available validation datasets have been produced in wind tunnels where 'low' *Re* is a constraint. To re-examine the effect of *Re*, we selected three reference wind velocities, namely 0.15 m·s$^{-1}$ (*Re* = 96,000), 1.5 m·s$^{-1}$ (*Re* = 960,000) and 5 m·s$^{-1}$ (*Re* = 3,200,000). The normalized average

magnitude of velocity ($U_{mean}/U_{ref}$) and dimensionless concentration ($C/C_{max}$) are averaged over a time-span of 300 $H/U_{ref}$, as shown in Figure 7, to employ equivalent mixing scales.

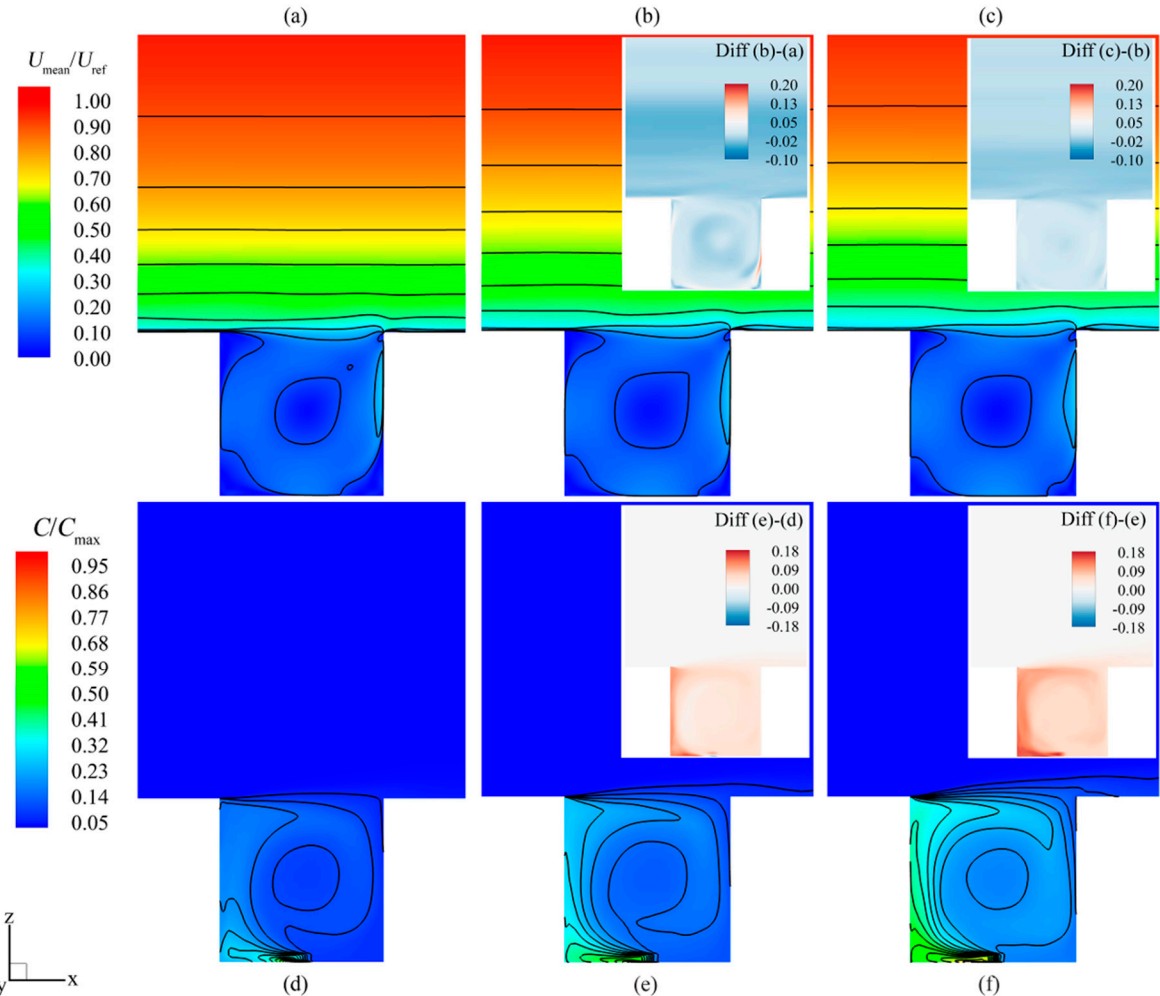

**Figure 7.** Normalized average velocity and concentration fields for the ideal street canyon ($AR = 1$): (**a**) and (**d**) at 0.15 m·s$^{-1}$, (**b**) and (**e**) at 1.5 m·s$^{-1}$, and (**c**) and (**f**) at 5 m·s$^{-1}$.

The similarity criterion is generally satisfied for the normalised velocity, as shown in Figure 7a–c, however, some differences also appear. In Figure 7b,c, the magnitude of the wind speed for the larger Re numbers at 960,000 and 3,200,000 is nearly identical, while for $Re = 96,000$ the height in which $U/U_{ref} = 0.5$ is lower than the former two. As $Re$ increases, the shear layer between the roofs is slight widening, while large vortex in the center of the canyon is reinforced, thus reducing the size of the three smaller ones. Moreover, the magnitude of velocity on the downwind wall is increasing along with velocity, meaning that the entrainment of clean air gets proportionally stronger with the increase of $Re$.

On the other hand, the impact on the pollutant dispersion is profound. The normalized concentration $C/C_{max}$ changes for the three $Re$ numbers, as presented in Figure 7d–f. The patterns of pollution are similar, but the frequency of the small concentrations changes drastically from $Re = 960,000$ and 3,200,000 to $Re = 96,000$ (Figure S2 of the Supplementary Material). The larger values of $C/C_{max}$ at $Re = 96,000$ are observed near the ground, at the height of the source. At $Re = 960,000$ and 3,200,000 pollution gets more mixed inside the canyon, with higher concentrations observed at the upwind side of the canyon and the upwind wall, respectively. Finally, it is noted that for $Re = 96,000$, $C/C_{max}$ values close to zero are simulated for the downwind side of the canyon.

The average normalized velocity for the three compared *Re* numbers showed discrepancies for the flow field, even though the notion of *Re* independency is generally confirmed. The results for concentration showed their clear dependency on *Re* number, as expected. Even though 3,200,000 is a high value of *Re* to simulate, the values for a real city geometry can be even higher, because of the large average height of the buildings. To keep the study as realistic as possible, the reference velocity at $5 \text{ m·s}^{-1}$ was chosen.

### 4.5. Source Height

A brief discussion regarding the pollutant source is presented. Having to simulate the effect of traffic pollution, we assume a line source, even though area sources have also been reported in the literature [94,95]. In OpenFOAM, the emission characteristics of the source are specified in the file *fvOptions*, inside the utility *scalarSemiImplicitSource*, while the topology is created with the *topoSet* utility [48].

The value for the mass emission rate was set to 620 g $\text{km}^{-1}\cdot\text{h}^{-1}$ and it represents a realistic emission rate of $NO_x$ for continuous canyon traffic [22]. Li et al. [69] claimed that the laminar layer formed on the walls creates a bottleneck to the dispersion of pollutants. This resistance creates non-physical results, when the volume of the source is located entirely inside the laminar layer of the wall. Thus, the height and width of the source should be investigated.

In the present study, the source is located at the middle of the canyon, so its width is set to two cells, which differs slightly depending on the grid (see Figure 1). Six potential source heights were examined, namely 0.005, 0.01, 0.1, 0.5, 0.75, and 1 m. The mass emission rate is kept steady for all source heights, by setting the selection *volumeMode* as *absolute* inside the utility *scalarSemiImplicitSource*. Figure 8 presents the normalized $C/C_{max}$ values after 900 s for three indicative source heights.

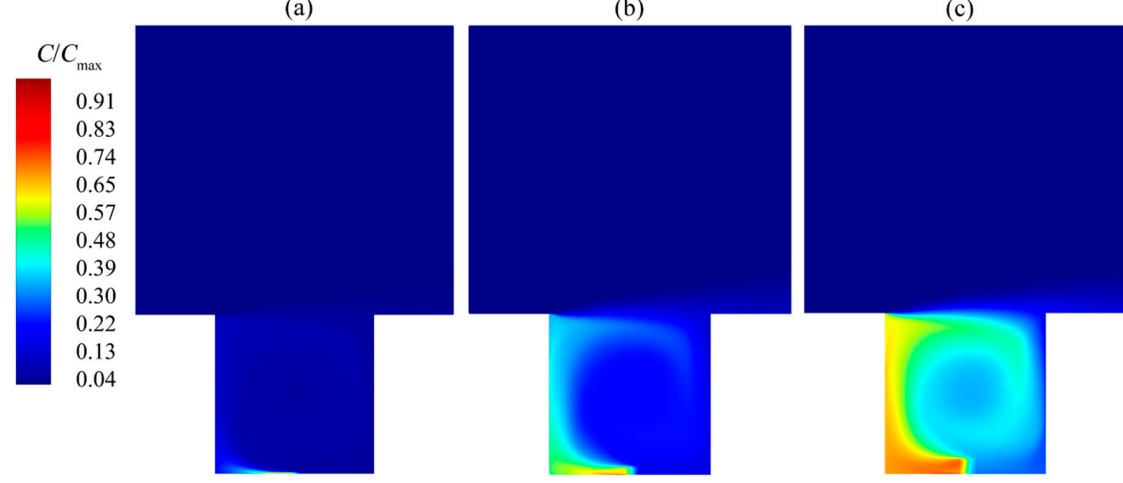

**Figure 8.** Average dimensionless concentration $C/C_{max}$ after 900 s for three source heights: (**a**) 0.1 m, (**b**) 0.5 m, and (**c**) 1 m.

Initially, the source height was set to 0.1 m above the ground that clearly resides in the wall boundary, as shown in Figure 8a. As a result, the pollutant is weakly dispersed, but gathered at the bottom of the street. Then, the source height was increased to 0.5 m (Figure 8b), resulting in a dispersion pattern similar to the experimental data, such as those by Pavageau and Schatzmann [31]. Finally, Figure 8c depicts the results of $C/C_{max}$ for the source height equal to 1 m, presenting a broader dispersion pattern. The results for 1 m improved the agreement with the experimental values for the downwind side, but increased the difference for the upwind side disproportionally. Thus, the source height of 0.5 m was selected as the optimal solution to both overcome the effect of the wall boundary layer and resemble the experimental data, even though the 1 m source height may also be considered a valid option.

## 4.6. Turbulence Modelling

Four SGS models (standard Smagorinsky, WALE, k-equation and localised dynamic k-equation) and three of the available wall functions (van Driest damping function, Spalding's law, and the simple logarithmic law) were tested in OpenFOAM v5.0. The considered scenarios are presented in Table 1.

**Table 1.** Combinations of the employed SGS models and wall functions.

| Combination | Abbreviation |
|---|---|
| standard Smagorinsky model and van Driest dumping function | S-VD |
| standard Smagorinsky model and Spalding wall function | S-SP |
| standard Smagorinsky model and *nutkWallFunction* | S-N |
| WALE model | WALE |
| k-equation model and van Driest dumping function | KE-VD |
| k-equation model and Spalding wall function | KE-SP |
| k-equation model and *nutkWallFunction* | KE-N |
| localized dynamic k-equation model | DKE |

The results of the simulations are shown in Figures 9 and 10, presenting the averages and fluctuations of *U*- and *W*-components, respectively. Three vertical lines at $x/W$ = 0.25, 0.50, and 0.75 of the canyon width were chosen for the comparison of numerical data obtained with the available experimental data by Li et al. [30].

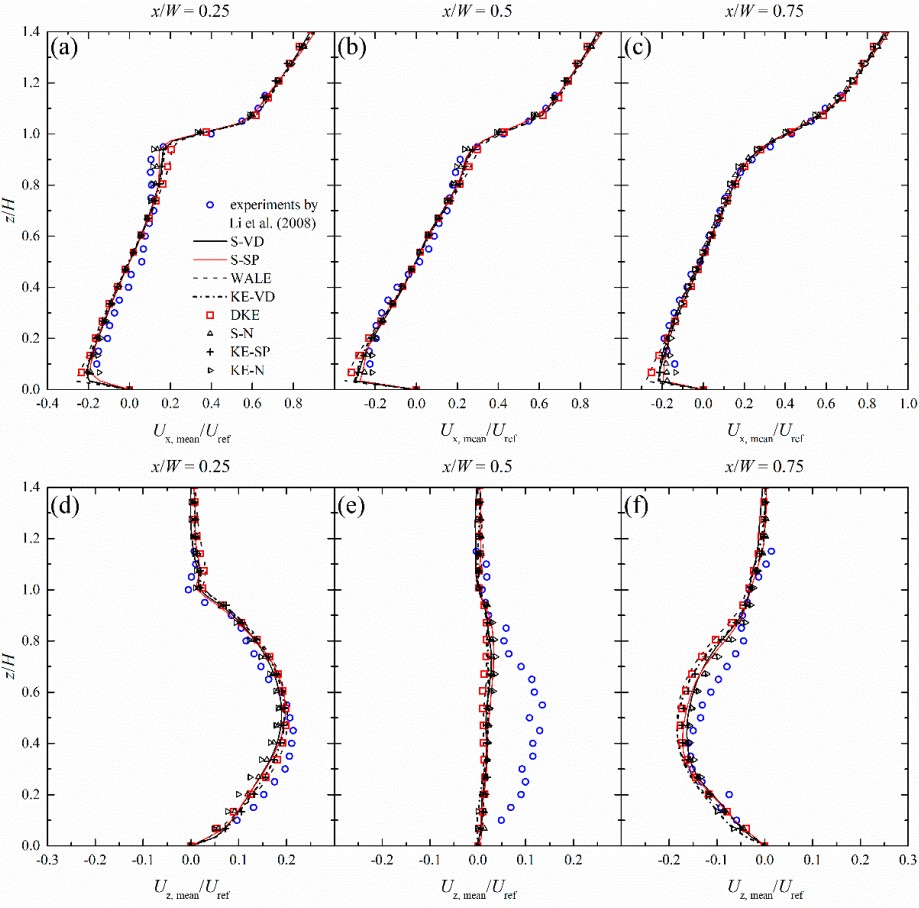

**Figure 9.** Normalized averages for the velocity components $U_x$ and $U_z$ for a unity street canyon at: (**a**) and (**d**) $x/W$ = 0.25; (**b**) and (**e**) $x/W$ = 0.50; and (**c**) and (**f**) $x/W$ = 0.75.

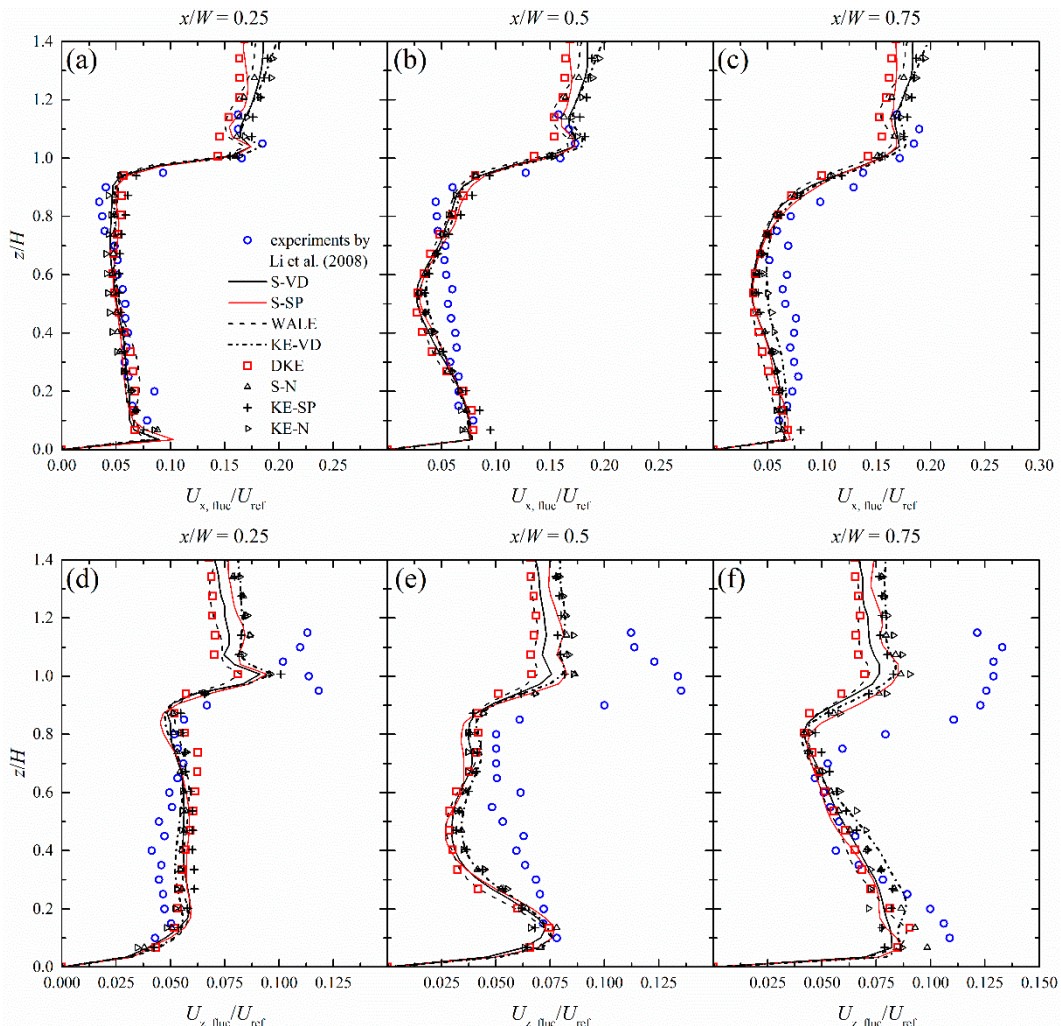

**Figure 10.** Normalized fluctuations for the velocity components $U_x$ and $U_z$ for a unity street canyon at: (**a**) and (**d**) $x/W = 0.25$; (**b**) and (**e**) $x/W = 0.50$; and (**c**) and (**f**) $x/W = 0.75$.

In Figure 9, there are no significant differences for the average $U_x$, between the applied models and the experimental data. The largest discrepancies are observed for $U_z$ in the middle of the canyon ($x/W = 0.50$), which applies to all the employed combinations. Mismatch from the experimental data is also observed for $U_z$ at $x/W = 0.75$, for the region between $z/H = 0.4$ and 0.8. For this case, the best prediction is obtained for KE-N, while the worst case represents WALE. Generally, the calculation for $U$ was proved to be more accurate than that of $W$.

This trend is also appeared for the velocity fluctuations, as shown in Figure 10. For this case, a disparity between the predictions and experiments appears as we move from $x/W = 0.25$ to 0.75. The capture of $U_{x,fluc}$ at $x/W = 0.25$ is in a good agreement with the available experimental data for all combinations, while for $U_{z,fluc}$ exists an over-prediction for the area between $z/H = 0.1$ and 0.7. For the line in the middle, the $U_{x,fluc}$ is over-predicted and under-predicted at the regions above and below $z/H = 0.7$, respectively, while $U_{z,fluc}$ is under-predicted for all the height. Finally, $U_{x,fluc}$ at $x/W = 0.75$ is generally under-estimated along the height of the canyon, but close to the experiments. On the other hand, the $U_{z,fluc}$ presents the worst discrepancies among the considered comparisons for the region higher than $z/H = 0.7$. In this area occurs the entrainment of clean air where the presence of turbulence is intensive and coincides with the area of lower $LES\_IQ_v$ (Figure 5b). The averages and fluctuations for $U_x$ and $U_z$ are compared with [30] in two more locations inside the canyon, at the Figures S3 and S4 of the supplementary material. For the predicted averaged $U_x$ and $U_z$ fluctuations, DKE model's

predictions present the largest mismatch compared to the experimental measurements, while KE-N model results are again approaching closer to the experimental data for all considered cases.

The most interesting observations obtained from the results of $C/C_{max}$, some of which are presented in Figure 2. The concentrations for the downwind wall are very close to each other and under-estimated for all the used combinations. For the upwind wall, the results differentiate a lot, with the largest over-prediction obtained from D-KE and WALE. Results obtained from S-V, S-SP, K-VD, and KE-N present almost identical performance and closer to the experimental but still resulting in over-prediction of the concentration field, while the last model exhibits slightly better performance. The $C/C_{max}$ results of all the employed SGS models and wall functions can be found in of the Figure S5 supplementary material.

K-SP and KE-N models exhibit the best predictions, as shown in Figure 2. Their results for the downwind side are in fair agreement with the experiments, while the K-SP model gives better predictions for the heights below $z/H = 0.4$. However, KE-N achieved the best improvement for the upwind side, out-performing the other combinations. Hence, the use of the k-equation model, together with the simple logarithmic *nutkWallFunction* (KE-N), is proposed herein.

*4.7. Validation Metrics*

The validation metrics *FAC*2, *NMSE*, and *FB* are employed to obtain a quantitative measure of the quality of the selected KE-N simulation, which was determined herein as the most suitable option for the prediction of the average concentration (in Section 4.6). The averages and the fluctuations of velocity components, at 200 points, were compared with the available experimental data by Li et al. [30], while the LES results by Li [73] were also used for comparison. The concentration results were validated against the experimental data by Pavageau and Schatzmann [31], Meroney et al. [32], Pavageau [33], and Hoydysh and Dabberdt [74]; for a total of 24 points. For the following definitions, $O_i$ are the observed values in experiments, and $P_i$ are the predicted values from the obtained LES, while "[ ]" denotes the averaged values.

1. Factor of two or *FAC*2

$$FAC2 = \frac{1}{N} \sum_{i=1}^{N} n_i \tag{23}$$

where $n_i = 1$, for $0.5 \leq P_i/O_i \leq 2$, or else $n_i = 0$.

2. Normalized mean square error (*NMSE*)

$$NMSE = \frac{\left[(O_i - P_i)^2\right]}{[O_i][P_i]} \tag{24}$$

3. Fractional bias (*FB*)

$$FB = \frac{[O_i] - [P_i]}{0.5([O_i] - [P_i])} \tag{25}$$

All the employed metrics are summarized in Table 2.

*FAC*2 estimates the percentage of the predictions, which are within a factor of two of the observations, based on the ratio of the predicted and observed value [75]. It was employed for $U_x$ and $U_z$ presenting a value of 0.85 for the horizontal velocity, the result for the vertical component is lower at 0.68 and the total *FAC*2 is 0.77. The ideal value is the unity, but values above 0.5 [77] and 0.8 [78] are also acceptable.

**Table 2.** Calculated and ideal values of the employed validation metrics and experimental datasets (200 points for the velocity components and 24 for the concentration).

| Metric | Quantity | | Result | Ideal |
|---|---|---|---|---|
| FAC2 | | $U/U_{\text{ref}}$ and $W/U_{\text{ref}}$ | 0.77 | 1 |
| | | $U/U_{\text{ref}}$ | 0.85 | 1 |
| | | $W/U_{\text{ref}}$ | 0.68 | 1 |
| NMSE | | $U/U_{\text{ref}}$ and $W/U_{\text{ref}}$ | 0.25 | 0 |
| | | $U/U_{\text{ref}}$ | 0.19 | 0 |
| | | $W/U_{\text{ref}}$ | 0.30 | 0 |
| FAC2 | upwind wall | $C/C_{\text{max}}$ | 0.82 | 1 |
| | | $C^*$ | 0.86 | 1 |
| | downwind wall | $C/C_{\text{max}}$ | 1.00 | 1 |
| | | $C^*$ | 0.90 | 1 |
| NMSE | upwind wall | $C/C_{\text{max}}$ | 0.05 | 0 |
| | | $C^*$ | 0.10 | 0 |
| | downwind wall | $C/C_{\text{max}}$ | 0.04 | 0 |
| | | $C^*$ | 0.26 | 0 |
| FB | upwind wall | $C/C_{\text{max}}$ | 0.06 | 0 |
| | | $C^*$ | 0.19 | 0 |
| | downwind wall | $C/C_{\text{max}}$ | 0.15 | 0 |
| | | $C^*$ | 0.10 | 0 |

Total NSME for both velocity components was found to be 0.25, while individually for $U_x$ and $U_z$ was calculated as 0.19 and 0.30, respectively. For this metric, the ideal value is zero, but *NMSE* < 1.5 [77] and *NMSE* < 4 [75] are likewise reported as acceptable limits. By these measures, the *NMSE* gives the higher level of confidence to our results. It is important to mention that the *NSME* does not have sense for parameters with negative and positive values [75], nevertheless, we tested it because 184 of the 200 points of our comparison have the same sign.

Regarding the concentration calculation, two ways are generally used. The first is the normalized concentration ($C^*$), employed to account for *Re* and strength of the source, described by Schatzmann and Leitl [96]

$$C* = \frac{C \cdot U_{\text{ref}} \cdot H}{Q/L} \tag{26}$$

where $C$ is the simulated concentration in Kg·m$^{-3}$, $U_{ref}$ is the reference wind velocity, $H$ is the characteristic building height and $Q/L$ is the strength of the source in Kg·m$^{-1}$·s$^{-1}$.

The second approach is the dimensionless concentration ($C/C_{\text{max}}$), where $C$ is divided by the maximum value in the domain, usually residing at the upwind wall. $C^*$ is used most of the times, even though the normalizing quantities vary, such as $H^2$ instead of $HL$ and volumetric flow of the source instead of mass rate [93]. In most of our cases, we applied the dimensionless concentration $C/C_{\text{max}}$, which was observed to reach a steady state value at around $500 \, H/U_{\text{ref}}$.

The three employed metrics for the dimensionless concentrations $C/C_{\text{max}}$ show excellent results. *FAC2* is 0.8 for the upwind wall, while it reaches the absolute unity for the downwind wall. The *NMSE* is very close to the optimal zero, presenting 0.05 and 0.04 for the upwind and downwind walls, respectively. Finally, *FB* gave the largest values at 0.15 for the downwind wall, while it was around 0.06 for the upwind wall that satisfies the recommended levels.

### 4.8. Turbulence Statistics

The energy spectra for the middle of the canyon were computed to examine the quality of our LES approach. Several works have utilized energy spectra to examine the ability of LES to reproduce realistic results. These are compared either against wind tunnel data [17,24], or among different LES parameterizations (e.g., methods for the production of turbulence [97], grid resolutions, and *Re* number [98]).

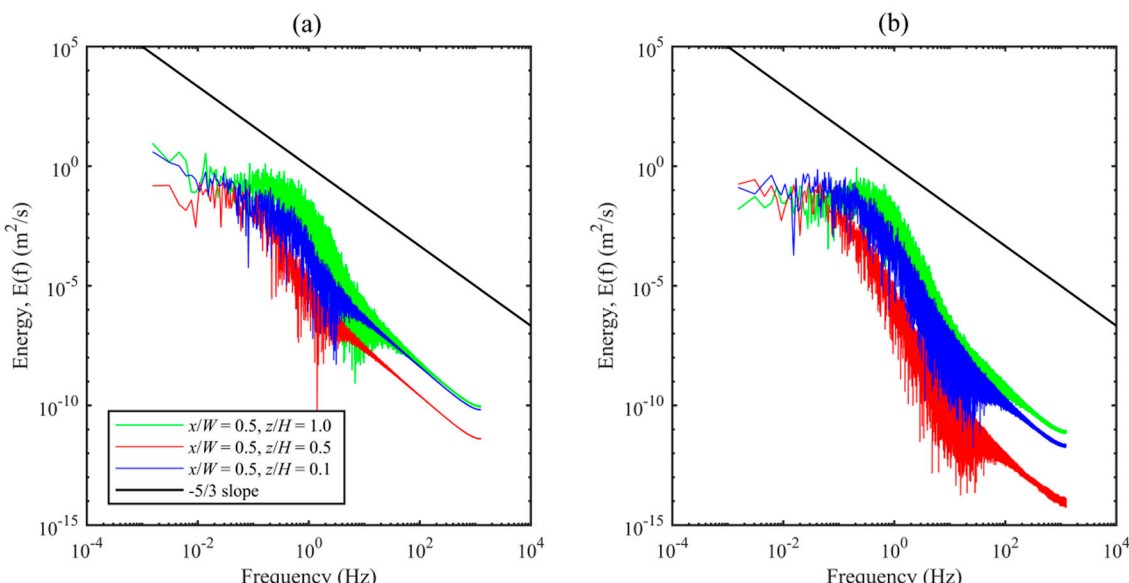

**Figure 11.** Turbulence energy spectra of the (**a**) $U_x$ and (**b**) $U_z$ velocity components, for three heights in the middle of the domain $x/W = 0.5$, at $z/H = 0.1$, 0.5, and 1.0.

Figure 11 presents the energy spectra for three heights in the middle of the canyon ($x/W = 0.5$), at $z/H = 0.1$, 0.5, and 1.0, where cells have relatively larger size compared to close the walls. For all the cases the amplitudes of the spectra correspond to the magnitude of the local velocity. The spectrum for $U_x$ at the top of the canyon is parallel to the 5/3 slope described by Kolmogorov, between 0.3 and 2 Hz, indicating the existence of the inertial sub-range in a narrow band. On the other hand, for the regions with the dense grid near the wall ($z/H = 0.1$) and the lower velocities in the middle height ($z/H = 0.5$), the spectra follow the $-5/3$ slope throughout the six orders of frequency, except from an area between 1 and 10 Hz. The spectra for $U_z$ are presented in Figure 11b and they show a very narrow inertial sub-range between 0.8 and 4 Hz. For higher frequencies and smaller turbulence scales the slope is steeper, implying that turbulence decays faster than expected. This may explain the inability of all combinations to predict $U_z$ in the middle of the canyon (see Figure 9) and can be attributed either on grid resolution or the numerical schemes.

## 5. Conclusions

This study focused on the LES of wind flow and pollutant dispersion in an ideal quasi-2D street canyon, using the open-source CFD code OpenFOAM v5.0. Initially, the computational domain height, the grid resolution and time-step were studied, to examine the quality of the LES approach. Moreover, the effect of *Re* and the height of the pollutant source were investigated, to find out their possible impact to the validation results. After establishing the details about the grid, the *Re* number and the source setup, the effect of turbulence modelling was analyzed. Four well-known SGS models that are implemented in OpenFOAM were tested, together with three standard wall-functions, to cover for the excess values of $y^+$. The results were compared with available experimental and numerical data. The main conclusions are:

1.  The height of the domain indicated a small effect for the concentration in the downwind side, while had no effect for the upwind wall. Both effects diminished for the case with 5 *H*, which was selected for the height of the upper atmosphere.
2.  The grid resolution with 120 cells per building height was selected. The use of the *LES_IQ*$_v$ index showed that turbulence kinetic energy was resolved at more than 80% for the area of the canyon, while the lower values at the street level are in the downwind side of the canyon.

3. The use of energy spectra showed small differences in the coverage of the turbulence scales for the examined time-steps, as expected, because the time-steps had very close values. Smaller values were not tested. A *CFL* value below 0.1 was kept for the most period of the numerical simulation and the largest part of the computational domain.

4. All the SGS models reproduced the average velocity at an acceptable level. Horizontal velocity $U_x$ is generally resolved better than vertical velocity $U_z$. Some discrepancies were observed for the fluctuations of velocity, again the larger differences were occurred for $U_z$.

5. On the other hand, pollutant dispersion was most affected by the SGS model, with best results obtained with the k-equation model and *nutKwallfunction*.

6. The height of the source can be critical to the dispersion of the pollutants and subsequently the validation process. If the selected height is located inside the laminar layer of the wall boundary layer on the ground, then dispersion will be artificially hindered. The selected source height overcame this issue and resulted to a better agreement with the experimental data. However, this is a known challenge of numerical models and a recurring issue in the literature, that requires better consideration in future experimental studies.

7. Three *Re* numbers were studied to examine its effect on the results. As expected, the normalized average velocity was nearly identical for the three cases. On the other hand, the average dimensionless concentration was affected and not proportionally to the *Re* number increase. The $y^+$ at the ground reached an average value of 2.7 and a maximum value of 6, while the respective values for the lateral building walls at 3.7 and 21. These values do not cover completely the proposed maximum of $y^+ = 5$. On the other hand, the use of the selected wall function and the results of the validation metrics imply that these $y^+$ values can be acceptable for *Re* = 3,200,000.

8. The discrepancies for $U_z$ in the middle of the canyon (Figure 9) can be attributed to the local grid resolution or the use of dissipative numerical schemes. In our case, second order schemes were applied for all terms, while the cells are relatively larger for this area. Furthermore, compared to the medium the fine grid showed a clear improvement, moving the slope of the energy spectrum for $U_z$, closer to the $-5/3$ slope described by Kolmogorov (Figure S6). This implies that a higher grid resolution may indeed improve the results in this region.

LES is a valuable tool and upon the selection and validation of the most appropriate configurations (e.g., turbulence schemes, grid resolution) could supplement and extended experimental work. Similarly to other modelling efforts, LES simulations could be used to enhance understanding of physical phenomena, and contribute on the development of new or improvement of existing numerical models and software. Such software could be capable to provide predictions or extensive hindcasts in almost real-time, thus their improvement could expand their applicability for emergencies.

**Supplementary Materials:** The following are available online at http://www.mdpi.com/2073-4433/10/1/17/s1. Figures S1–S6, a sample animation (Figure S7), and the OpenFOAM Source files (without the calculated mesh or initial time step).

**Author Contributions:** Conceptualization, K.E.K.; Methodology, A.E.C., C.D.A. and K.E.K.; Software, Validation, A.E.C. and C.D.A.; Formal Analysis, Investigation, Data curation, Writing—Original Draft Preparation, A.E.C.; Writing—Review and Editing, all authors; Supervision, Project Administration, Funding Acquisition, K.E.K. and M.J.A.

**Funding:** This publication was made possible by a NPRP award (NPRP 7-674-2-252) from the Qatar National Research Fund (a member of The Qatar Foundation). The statements made herein are solely the responsibility of the authors.

**Acknowledgments:** The HPC resources and services used in this work were provided by the IT Research Computing group in Texas A&M University at Qatar. IT Research Computing is funded by the Qatar Foundation for Education, Science and Community Development (http://www.qf.org.qa).

**Conflicts of Interest:** The authors declare no conflict of interest.

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
