# Peer review of "Qualitative and Quantitative Investigation of Multiple Large Eddy Simulation Aspects for Pollutant Dispersion in Street Canyons Using OpenFOAM"

_atmosphere, doi:10.3390/atmos10010017_

Round 1
Reviewer 1 Report
Overview:
The present article performs dispersion simulations of an idealized street canyon by using Large Eddy Simulations. The authors analyzed the most common set-up decisions that can make the difference between success and failure when comparing the results with wind tunnel data. The study is well organized, and includes figures that are adequate for the research. The authors guide the readers across the different set-up characteristics, defining at the end of the article the best configuration compared to the experimental data for their case.
Main drawbacks: lack of originality, since similar studies for LES have been performed. The idealism of the case, which seems not to need a turbulence generator, which is one of the most complex configurations dilemma nowadays in LES.
Main strong points: clear guidelines for running LES with open foam, which can serve as a starting point for plenty of researchers using this open source tool nowadays.
Missing sections/details suggestions:
There is not a section in the article specifying how the authors are generating turbulence. Are the authors just running long enough for turbulence to developed by itself? this should be explained more in detail somewhere in the text.
In the results section, it is not very clear why KE-N model outperforms the dynamical SGS model, or other models. More hypothesis regarding this would be valuable for future studies.
Through out the article it is not really clear to the reader if the simulations are performed in a real scale canyon or a wind tunnel size. Could the authors specifically include this in the text?
Specific suggestions/doubts:
l.43: “urban turbulent flows can be obtained”, considering that urban turbulent flows at real scale can not be solved by DNS, I would suggest the authors to modify by “Solutions of the Navier-Stokes equations can be obtained…”.
l. 56: “is more accurate than URANS, is the LES”. LES is indeed more accurate in resolving turbulence, and therefore representing better the effects of it within the flow. However, LES applied to real scale atmospheric flows (not controlled environments) may face the same problems as RANS regarding boundary conditions and variability. I would suggest to rephrase to “represents turbulence more accurately than URANS, is the LES”.
l.64-71: the authors introduce a set of references supporting the choice of LES, however all those studies show the outperformance of LES compared to other models for wind tunnel studies. More references showing LES applied to real scale urban flows and compared with field experiments may provide a more general and fair view of the LES capabilities.
l.73-79: the authors summarized the work of Chung and Liu [25], however it is not really well stated the point of the paragraph towards the present study, or any specific conclusion. Please rephrase for clarity or remove that paragraph.
l.107-109: Is this claim supported by the authors? How would high resolution LES replace field experiments for the dispersion in cities? I would suggest the authors to be more specific referring to wind tunnel experiments, where a controlled environment is assumed.
l.132: “wind direction (left to right)”, the authors could add an arrow in figure 1, which would be more intuitive for the reader.
l. 267-268: is the diverging problem only occurring in v3.0? could the authors be more clear on this?
l.299: “Solutions” -> solution.
l. l349-351: the authors used the coarse mesh (most dissipative) to verify if different domain heights might affect the results. Would this conclusion change by using the finest mesh?
l.394: “close to the grid-independent solution”, is there grid independent solution in LES if your SGS depends on the resolution of the mesh?
l.526: is the injection number of cells different across meshes? and if it is, wouldn’t this affect the injection rate, and create differences between the simulations when normalizing only by the concentration and not the volume of injection?
Figures suggestions:
Figure1: I do not really understand in (a) what are the discontinuous line or the shaded areas. Could these be explained in the caption of the figure, or within the text, or in the figure with arrows or all at once?
Figure7: Why the authors compared b-a, e-d, and then c-b, f-e? Would it be more clear to compare always with the same reference to see the change across different velocities? c-a and f-d.
Figure9: “Li et al. 2008” corresponds to experiments? if so, it should be indicated in the legend, since other results from Li corresponding to numerical predictions have been shown previously.
Equations suggestions:
Ce in the text is represented as C\varepsilon in the equations, could the authors choose one of the two, and modify accordingly?
Author Response
The reviewers’ comments and suggestions were very helpful and improved our manuscript. We have addressed all the concerns, while all the corrections and modifications are presented in the revised manuscript with the Track Changes coloring. Minor grammatical and syntax errors were corrected in the whole paper. Our answers have been provided in the attached file and in red color

Reviewer 2 Report
This paper investigated a variety of key parameters (including computational domain height, grid resolution, time step,Reynolds number, Source height and turbulence SGS models, wall functions) in the large-eddy simulation of pollution dispersion for an idealised 2D street canyon. The models were well compared with some measurements and numerical models in the literature. This paper is well structured and well-written so minor comments are suggested before publication:
1) Lines 662-663 may add other aspects that presented in the results (e.g. Reynolds number, Source height).
2) Line 677 may also add some major results (e.g. Reynolds number, Source height) into the main conclusions (Line 666).
3) Line 680, As k-equation model and nutKwallfunction performed best for pollutant, why it is not chosen for the test of computational domain height, grid resolution, time step,Reynolds number and Source height. Is the standard Smagorinsky model and van Driest wall function used for these? Could explain more about this?
4) Any future direction/recommendation could be added in the end of the conclusion section?
Author Response

(The authors gave the same response as above.)
